# Increased General, Eating, and Body-Related Psychopathology in Inpatients in a Specialized Eating Disorders Unit after the Beginning of the COVID-19 Pandemic: A Retrospective Comparison with the Pre-Pandemic Period

**DOI:** 10.3390/jcm12020573

**Published:** 2023-01-10

**Authors:** Matteo Martini, Paola Longo, Nadia Delsedime, Giovanni Abbate-Daga, Matteo Panero

**Affiliations:** Eating Disorders Center, Department of Neuroscience “Rita Levi Montalcini”, University of Turin, Via Cherasco 11, 10126 Turin, Italy

**Keywords:** anorexia nervosa, hospitalization, COVID-19 pandemic, eating disorders, body-related symptoms, lockdown, mental health

## Abstract

The study of the effects of the COVID-19 pandemic on individuals who attended mental health services is needed to identify the specific vulnerabilities associated to this challenging period. Despite several analyses reporting the worsening of eating disorders symptomatology after the beginning of the pandemic, characterizations of adult inpatients with eating disorders are still lacking. We conducted a retrospective analysis to assess whether adult individuals who underwent hospitalization in a specialized eating disorders unit in the two years after the beginning of the COVID-19 pandemic differed in clinical presentation, psychopathological measures, and treatment outcomes from inpatients hospitalized in the two years before. In the comparison between the two groups, the individuals who began treatment after the start of the pandemic presented with more physical hyperactivity and more severe psychopathological scores in most of the areas investigated, with differences in eating symptoms still evident at discharge. Notably, body-related symptoms (i.e., body shape concerns, body checking, body avoidance) were associated with the pandemic, and also for inpatients with extreme anorexia nervosa. This retrospective analysis does not allow us to separate the impact of COVID-19 from other potentially relevant co-occurring factors; however, these findings help in understanding how the pandemic could have affected individuals that needed specialized intensive treatment.

## 1. Introduction

Soon after its inception in 2020, the COVID-19 pandemic was recognized to affect psychological well-being [1]. The increase in psychiatric symptoms was documented to be greater for those individuals with pre-existing mental disorders compared to the general population [2,3]. In these groups, individuals of younger age seemed to be more impacted by the pandemic in comparison to older people [4], probably due to the diminishing of social contact, which is crucial for socialization in childhood and adolescence.

Studies on individuals with eating disorders (EDs) generally showed worsening symptoms during the pandemic [5,6,7,8,9,10,11], with analyses made both on different symptomatologic domains such as general, eating, and trauma-related psychopathology [12], and at different time points (e.g., lockdown and re-opening phases). For instance, Castellini and colleagues [13] examined outpatients with an ED both before the pandemic and in the first months after its inception and compared them to healthy controls, finding increased compensatory exercise in patients with an ED and re-exacerbation of binge symptoms, as well as in remitted individuals. Monteleone and colleagues [14] documented a worsening of both general and eating psychopathology during lockdown periods that persisted also during re-opening phases for most of the dimensions investigated. Notably, in this study, anxiety symptoms increased not only in the lockdown phase in comparison to non-pandemic times but also during the re-opening in comparison to the lockdown phase. Anxiety is indeed a transdiagnostic prominent feature of ED psychopathology [15]. The combination of the heightened interpersonal sensitivity of individuals with EDs with the reduction of social contacts and the increase in media exposure during lockdown (with media content reporting COVID-19 potential dangers to personal safety, but also amplifying body image concerns such as the risk of becoming fat due to lack of exercise) could have resulted in a further increase in anxiety after the end of the lockdown period.

Despite the studies reporting the worsening of ED psychopathology after the beginning of the COVID-19 pandemic, the evidence on the presentation to specialized ED treatments and clinical outcomes in this period is still mixed. For instance, the systematic review by Sideli and colleagues [16] found the worsening of symptoms to occur more often in individuals with self-reported ED than in those attending ED services. Carr and colleagues [17] compared online treatment after the beginning of the pandemic to the previously delivered face-to-face day service treatment and found it effective in decreasing EDE-Q scores but not for weight improvement. These data point to the need for a better characterization of individuals with ED who attended specialized services during the pandemic, in order to evidence both differential (e.g., with respect to either the pre-existence or the severity of the disorder) areas of vulnerability and protective factors connected to specific aspects of the treatment delivered.

Regarding inpatient ED treatment, an increase in the percentage of hospitalizations has been reported in children [18] and adolescents [19], whereas the hospitalization rate for this age group was generally stable in the 5 years before the pandemic [20]. The systematic review by Devoe and colleagues [21] documented an over 80% increase in pediatric admissions and a 16% increase in adult admissions. In this review, ED symptoms, anxiety, and depressive symptoms were found to be increased, whereas mixed results emerged regarding BMI changes after the beginning of the pandemic. Waiting time from referral to hospital admission increased from 33 to 46 days at the beginning of the pandemic in the study by Ayton and colleagues [22].

Most of the retrospective studies on hospitalization were conducted on children and adolescent populations [20,23,24]; therefore, still few data are present in the literature on adult inpatients [22]. Very recently, Schreyer and colleagues analysed data on adults (mean age around 30 years old) and adolescents (mean age around 15 years old) inpatients with EDs and compared them to pre-pandemic cohorts, finding an effect of COVID-19 on eating and depressive symptoms only for the adolescent group [25]. For both age groups, no significant differences were found regarding anxiety levels at admission and weight outcome at discharge. This study relevantly addresses the greater vulnerability of younger individuals to the pandemic; however, there are still gaps to cover in order to understand the specificities of the effect of COVID-19 on inpatients with severe ED. For instance, a characterization of body-related concerns in this population after the start of the pandemic is missing. Previous work demonstrated that body-related concerns are relevant in the context of ED hospitalization and closely related to treatment outcomes [26]. Furthermore, it would be useful to assess whether individuals hospitalized for extreme-AN (i.e., with a Body Mass Index (BMI) inferior to 15 kg/m^2^) after the start of the pandemic presented a similar or different patterns of symptoms in comparison to the pre-pandemic cohort [27,28].

In summary, the literature here reviewed demonstrates that studies on clinical presentation at admission and outcome at discharge from adult ED inpatient treatments during the COVID-19 pandemic are still scarce. Furthermore, a characterization of specific psychopathological areas that prior to this period were identified as relevant for this population (i.e., body-related symptoms) is still missing.

### Aims

The first aim of this retrospective analysis was to compare adult inpatients with an ED who underwent hospitalization during the two years after the beginning of the COVID-19 pandemic to those treated during the two years before the pandemic on (a) presentation at admission and treatment outcome, (b) self-reported eating, depression and anxiety measures, (c) specific measures on body checking, avoidance and body shape concerns. Secondary aims of the analysis were (a) to compare clinical presentation at admission during lockdown periods and during re-opening phases, and (b) to offer a characterization of inpatients with extreme-AN before and after the start of the pandemic. Aim (a) and (b) have been explored in the adult ED inpatient population by Schreyer et al. [25] albeit with a different focus (i.e., comparison between adolescent and adult populations), whereas aim (c) is novel. Similarly, the secondary aim (a) has been explored in the ED population attending specialized ED centers [14], whereas we are not aware of studies assessing the secondary aim (b). We expected to find a generalized worsening of ED symptoms after the beginning of the pandemic. Due to current literature limitations, we had no prediction regarding specific differences in the groups of individuals with extreme-AN.

## 2. Materials and Methods

Participants were recruited among individuals with the diagnosis of an ED who were hospitalized at the inpatient unit of the Eating Disorder Center of the University of Turin and completed clinical evaluation and self-reported questionnaires on eating and general psychopathology at admission and discharge. The period considered for this retrospective analysis was the two years before and after March 2020. The ED Center is a secondary treatment center fully founded by the public Italian healthcare system, and access to care is free (i.e., not based on insurances). All individuals treated in the period considered were non-Hispanic White.

Exclusion criteria were: (a) age < 18 and >65 years old (i.e., adult population); (b) presence of psychotic, bipolar, or active substance use disorders; (c) incomplete baseline questionnaires on eating, anxiety, and depression symptomatology.

Out of 212 total admissions recorded in our databases for the period considered, a total of 159 fulfilled the criteria above (80 for the two years before and 79 for the two years after March 2020). A total of 53 individuals were excluded from the analysis because the clinical or psychometric data at admission were incomplete or missing (e.g., due to the patient’s refusal to fill out admission questionnaires).

A total of 12 individuals included in the post-pandemic group were also present in the pre-pandemic group. Re-running the analyses after excluding them did not alter the results (Appendix A); therefore, data on the whole sample are presented.

For a total of 99 individuals (53 pre-pandemic; 46 post-pandemic), questionnaires on body-related measures were available.

### 2.1. Procedure

All patients admitted to our unit are evaluated by an expert psychiatrist and diagnoses are made following the Structured Clinical Interview for DSM-5 [29]. Height and weight are measured by trained nurses at admission to calculate BMI. Weight is periodically measured during the stay and before discharge from the unit. Admission to our unit often happens in emergency conditions (e.g., directly after access to the emergency room) and the first aim of the treatment is medical stabilization. As suggested by international guidelines, treatment is provided by a multidisciplinary team and individual daily structured sessions are offered to the patient with the aim to enhance treatment adherence, have a psychological understanding of the causes of symptoms worsening, and foster motivation to engage in further steps of the treatment. Self-reported questionnaires are administered in the first ten days after treatment admission, and in the last week before treatment discharge.

Written consent is acquired and routine assessments are run according to the principles of the declaration of Helsinki. The study was approved by the hospital Ethical committee (approval number 0036472).

### 2.2. Measures

For this analysis, clinical measures considered were diagnosis, length of stay, duration of illness, presence of binge-purging symptoms, presence of excessive physical activity, BMI, and caloric intake at admission and discharge. Clinical measures were collected via a form filled out by the clinician after the admission visit and at discharge. Duration of illness was either known for individuals who attended the Center in other settings or reconstructed via patient history collection. Similarly, the presence of binge-purging symptoms and excessive physical hyperactivity were both operationalized as dichotomous variables, indicating whether these symptoms were present at a clinical level in the month prior the assessment at admission; their presence during the inpatient stay was assessed via daily clinical observation and visits. Caloric intake at admission was estimated through self-report with the help of the nutritionist that then followed the patient during the hospital stay. Symptoms of social anxiety at the beginning and end of treatment were measured through the administration of the Brief Social Phobia Scale (BSPS) interview [30]. Eating symptomatology was assessed with the Eating Disorders Examination Questionnaire (EDE-Q) completed both at admission and discharge [31]. Cronbach’s alpha in the sample was 0.95.

Anxious and depressive symptoms at admission were investigated with the State and Trait Anxiety Inventory (STAI) for anxiety [32], and Beck Depression Inventory (BDI) for depressive symptomatology [33]. Cronbach’s alpha in the sample was 0.96 for STAI and 0.87 for BDI.

In the subsample for which these questionnaires were available, body-related psychopathology at admission was examined through the Body Image Avoidance Questionnaire (BIAQ) [34], the Body Checking Questionnaire (BCQ) [35], and the Body Shape Questionnaire (BSQ) [36]. Cronbach’s alpha in the sample was 0.86 for BIAQ, 0.96 for BCQ, and 0.97 for BSQ,

### 2.3. Data Analysis

Firstly, inpatients hospitalized in the two years after the start of the pandemic were compared to those hospitalized in the two previous years on clinical and psychopathological measures at admission and discharge. Then, we conducted a comparison between the two groups on body-related variables at admission for those individuals for which body-specific questionnaires were available.

We then conducted the same analyses on two subpopulations, namely (1) the pandemic group only (i.e., the individuals hospitalized after March 2020), by splitting it in lockdown and re-opening phases, and (2) the group of individuals with extreme-AN (i.e., with BMI < 15 kg/m^2^). For the first of these two analyses, individuals in the pandemic group were divided into those who began hospital treatment during lockdown periods and those who were hospitalized during the re-opening phases that occurred between the lockdowns. Lockdowns in Italy entailed nationwide quarantine and the closure of all non-essential business and were put in place in the following periods: first lockdown between March 2020 and June 2020; second lockdown between October 2020 and the end of April 2021; a phase of movement restrictions (i.e., the enduring of the state of emergency with limitations in access to non-essential business) between December 2021 and the end of March 2022. For the second of these analyses, we compared individuals with a BMI < 15 kg/m^2^ hospitalized in the two years before March 2020 to those inpatients in the same BMI range hospitalized in the two years after March 2020.

Finally, we conducted a logistic regression with the variables that resulted significantly in the multiple comparisons in the whole sample to identify a simpler model that explained the differences between the pre-pandemic and the pandemic inpatient groups. The distribution of the variables was inspected graphically and with the use of the Shapiro-Wilk test. Two Sample *t*-test, Fisher’s exact test, and Pearson’s Chi-squared test were run where appropriate. Imbalances in diagnostic subgroups pre and post-pandemic were investigated with the use of factorial ANOVAs. To account for alpha inflation, Holm-Bonferroni correction was applied for each group of comparisons. Since there are controversies in the literature around the use of multiple comparison correction—especially when multiple results in the analysis are consistent with each other—we report both uncorrected and corrected *p*-values. Effect sizes were measured with Cohen’s d and interpreted following the common rule of thumb (small effects for d = 0.2, medium for d = 0.5, and large for d = 0.8). To build the logistic regression model, multicollinearity was assessed with variance inflation factors (VIF). Tjur’s R2 was used to assess the model’s explanatory power [37]. All analyses were run in R using RStudio.

## 3. Results

### 3.1. Clinical and Psychopathological Measures at Admission and Discharge

Regarding information on symptoms collected by clinicians, inpatients hospitalized after March 2020 showed more frequently excessive physical activity, both at admission and discharge, and higher physiological symptoms of social anxiety—as measured by the BSPS interview—at admission and discharge compared to those hospitalized before the pandemic (Table 1). BMI did not differ between the two samples. After applying a multiple comparison correction, only excessive physical activity at admission remained significant. Regarding symptoms questionnaires, inpatients hospitalized after March 2020 scored significantly higher in all EDE-Q, STAI, and BDI scales at admission. Similarly, all EDE-Q subscales were significantly higher at discharge for this group of patients (Table 1). The delta in EDE-Q global scores did not differ in the two samples (t(89.59) = −0.54, *p* = 0.592). After correcting for multiple comparisons, differences at admission remained significant for EDE-Q restraint, shape concern, weight concern, global score, STAI state and trait anxiety, and for all EDE-Q subscales at discharge. Most effect sizes were of medium entity (i.e., Cohen’s d around 0.5).

The group of individuals hospitalized after March 2020 showed an increase in percentage of individuals diagnosed with AN-BP (Table 1), even though there were no significant differences in the comparison of all the diagnoses. To account for potential effects of subgroup imbalances, we conducted factorial ANOVAs on the population diagnosed with AN, using the presence of hyperactivity, EDE-Q and STAI subscales as dependent variables, and the period of hospitalization (before and after COVID-19), the diagnostic sub-type (AN-R and AN-BP), and the interaction between the two as fixed factors (Appendix A). For all variables considered there was an effect of the period of hospitalization, the post-COVID period being associated with significantly higher scores. For EDE-Q restraint, eating concern, weight concern and global score there was also an effect of diagnostic subtype, AN-BP being associated with higher scores. For no variables there was a significant effect of the interaction between the period of hospitalization and diagnostic subtype.

### 3.2. Body Checking, Avoidance, and Shape Concerns

Regarding the subgroup of inpatients for which questionnaires on body checking, avoidance, and shape concerns were available (N = 99), inpatients hospitalized after March 2020 showed significantly higher scores on all dimensions except for BIAQ grooming/weighing subscale and all differences remained significant after applying the Holm-Bonferroni correction (Table 2). Effect sizes were medium to large.

### 3.3. Lockdown and Re-Opening Comparison

The comparison between inpatients hospitalized during lockdowns and during the re-opening phases did not show differences in BMI nor EDE-Q subscales. Initially, significantly higher scores in physiological symptoms of social anxiety reported in the BSPS interview emerged at discharge for the re-opening group (Appendix A); however, after controlling for multiple comparisons, no significant differences survived. Figure 1 shows BMI (for individuals underweight at admission) and EDE-Q global scores at admission and discharge for inpatients hospitalized before and after March 2020 and for inpatients hospitalized during and after lockdown periods.

### 3.4. Extreme-AN

Regarding general clinical measures and questionnaires on the 74 inpatients with extreme-AN (34 pre-COVID, 39 post-COVID), individuals with extreme-AN after March 2020 presented at admission a significantly higher frequency of excessive physical activity, and significantly higher scores in all psychopathological questionnaires. After accounting for multiple comparisons, EDE-Q subscales and STAI trait subscale remained significant (Appendix A). Regarding specific body questionnaires, inpatients after March 2020 showed significantly higher scores in all subscales except for BIAQ clothing, and—after multiple comparison correction—BIAQ eating-related control behavior. Effect sizes ranged from medium to large. Figure 2 shows total scores in BIAQ, BCQ, and BSQ for inpatients with extreme-AN hospitalized before and after March 2020.

### 3.5. Logistic Regression

The variables at admission that resulted in being significant in the comparison of the two whole samples were inserted as predictors in a model that had the binary outcome of being part of the pre-pandemic or post-pandemic group. Therefore, we estimated a model using the presence of physical hyperactivity, EDE-Q restraint, shape concern, weight concern, STAI state and trait subscales. In this model, EDE-Q shape concern and EDE-Q weight concern demonstrated a moderate level of collinearity (i.e., VIF around 5); therefore we excluded the EDE-Q weight concern. The final model had a moderate explanatory power (Tjur’s R2 = 0.17), with a significant and positive effect of the EDE-Q shape concern in predicting the probability of being in the post-pandemic group (Table 3).

## 4. Discussion

In this retrospective study, we analyzed clinical and psychopathological data from adult inpatients who underwent hospitalization in a specialized ED unit in the two years after the beginning of the COVID-19 pandemic and compared them to inpatients hospitalized in the two years before. In comparison to the pre-pandemic group, patients hospitalized after the beginning of the pandemic did not differ in mean BMI; however, they presented at admission with more frequent excessive physical activity, increased anxiety symptoms, and higher levels of eating psychopathology. Notably, the difference in eating psychopathology remained evident at discharge, with the two groups showing a comparable reduction in eating symptomatology during the inpatient stay. Furthermore, the pandemic group was characterized by higher body concerns, a finding that emerged both from the comparison of questionnaires specifically assessing body checking, body avoidance, and body shape concerns, and by the results of a logistic regression model that evidenced EDE-Q shape concerns as the dimension specifically tied to this group. Differences in specific eating and body-related psychopathology were also evident for the subgroup of inpatients with extreme-AN (i.e., with a BMI < 15 kg/m^2^). No differences were found between inpatients hospitalized during lockdowns and re-opening phases.

Our results are in line with the literature reporting worsening of specific and general psychopathology in individuals with EDs during the pandemic [14,16] and pinpoint body-related concerns as clearly associated with this period of time for those individuals that needed a specialized ED inpatient treatment. Our results seem to differ from those of Schreyer and colleagues [25], which did not evidence an effect of COVID-19 on eating symptoms in adult inpatients. This could be partly due to the fact that the mean age of our adult sample was lower (i.e., mean age around 25 years old) in comparison to the study by Schreyer et al. (i.e., mean age around 30 years old). These results could suggest that younger adults seem to have been affected by the pandemic similarly to adolescents.

Regarding clinical measures, the present analysis suggests that the pandemic was associated with more frequent excessive physical activity. These findings are in line with the increased levels of compensatory exercise documented in outpatients with EDs at the beginning of the pandemic [13]. Our results suggest that this phenomenon was not transient but endured in the following period, similar to what has emerged for anxiety symptoms in the study by Monteleone et al. [14]. Regarding this dimension, also in our sample anxiety symptoms significantly increased during the two pandemic years considered. Furthermore, clinicians-rated physical symptoms of social anxiety also seemed to be higher both at admission and at discharge from treatment, even though this finding should be taken cautiously since it did not survive correction for multiple comparisons. Interestingly, both hyperactivity and bodily manifestations of social anxiety can be easily put in relation to the emergency measures adopted during lockdowns to tackle COVID-19, which entailed restriction of movements and reduction of social contacts. It can be hypothesized that periods of confinement associated with limited external social interactions heightened both the urge to engage in maladaptive physical activity and the fear response to social contact in this vulnerable population. This would be consistent with the conceptualization of hyperactivity and pathological eating behaviors as coping mechanisms for excessive anxiety in individuals with EDs [15]. Further studies may deepen the impact of the pandemic on social and emotional skills in patients with ED, given the significant role of these skills in AN [38].

Eating symptoms measured with the EDE-Q were higher both at baseline and at the end of treatment in the pandemic group in comparison to the pre-pandemic group, even though both groups showed a similar decrease in symptoms during hospitalization. BMI at presentation and discharge for the underweight ED population, however, did not differ before and after the pandemic, and the two groups showed similar weight improvement during hospitalization, in line with some previous studies [21]. It is interesting, however, that worsening in cognitive severity of the disorder particularly regarded those psychopathological dimensions (i.e., overvaluation of weight and shape) that are considered central to ED pathology by current explicative models. Furthermore, in comparison to the two years before, not only body shape concerns but also body-checking and body image avoidance were reported as higher and these differences were particularly evident for those with lower BMI. These findings could be related both to the specificity of the disorder and of the pandemic phase. We could speculate that individuals with acute and severe EDs have been particularly prone to worsening of the rumination on body image during lockdown—maybe also in connection to increased media exposure—which, in turn, entailed the worsening of behavioral symptoms such as body checking.

The separate analysis of individuals with extreme-AN also suggested that this population was affected by the pandemic. Individuals with extreme-AN hospitalized after March 2020 showed higher levels of psychopathology in many of the investigated areas in comparison to the pre-pandemic group of extreme-AN. This finding could be interpreted in the sense that having reached an extremely low BMI was not linked to either minimization or actual reduction of pathological concerns [27,28].

This study is one of the first to provide a characterization of several clinical and psychopathological variables of adult ED inpatients during the COVID-19 pandemic and has the strength of documenting how core ED symptoms (i.e., body-related concerns) were impacted during these challenging times in individuals with acute and severe EDs. Some limitations, however, should be acknowledged. The retrospective design warrants caution in interpreting the findings. Notably, we could not isolate the impact of the COVID-19 pandemic from other factors that might have contributed to the differences found between the two separate samples. On this line, we cannot infer data on ED incidence, nor on an increased rate of ED individuals in need of hospitalizations in our region. Even if our outpatient service continued to function during the pandemic, the reduction of in-person visits could have contributed to a less prompt response to clinical worsening and therefore to more severe presentations. The presence of hyperactivity was evaluated by the clinicians; however, it was considered as a dichotomous variable (i.e., clinically present or absent), thus limiting more precise analyses on the time spent per day doing physical exercise. The pandemic caused some disruption in data collection. The number of individuals in the subgroup analyses was limited, especially for individuals with extreme-AN. The findings of the present study were obtained from White individuals, and therefore lacking generalizability to individuals of more diverse racial and ethnic origin. Furthermore, we did not report information on sexual orientation and gender identity since it was not consistently collected.

In conclusion, the present study showed that young adults with an ED who were hospitalized in the two years following the beginning of the COVID-19 pandemic presented at admission and discharge more severe psychopathological symptoms in comparison to individuals hospitalized in the two years before. The more striking differences between the two groups concerned the presence of excessive physical activity and body-shape related concerns, which were also evident for individuals with extreme-AN. Future follow-up studies may clarify whether the pandemic induced long-lasting modifications in ED presentation or simply accelerated phenomena that were present before its inception.

## Figures and Tables

**Figure 1 jcm-12-00573-f001:**
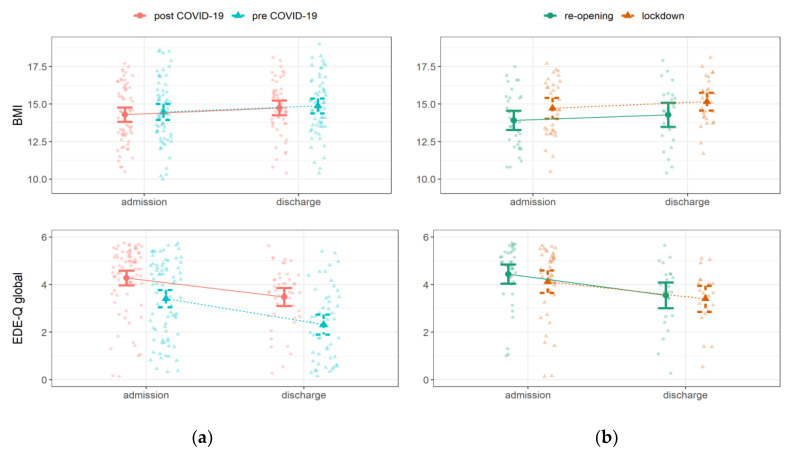
Changes in Body Mass Index (BMI) and Eating Disorder Examination Questionnaire (EDE-Q) during inpatient stay for the (**a**) whole sample and for (**b**) the pandemic group.

**Figure 2 jcm-12-00573-f002:**
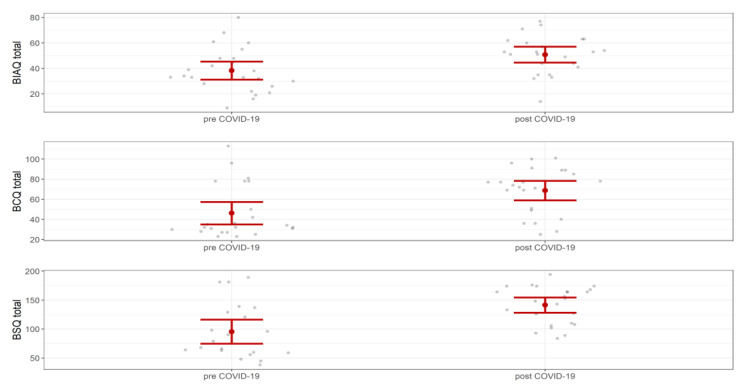
Body Image Avoidance Questionnaire (BIAQ), Body Checking Questionnaire (BCQ), Body Shape Questionnaire (BSQ) total scores in inpatients with extreme-AN.

**Table 1 jcm-12-00573-t001:** Comparison of clinical and psychopathological measures at admission and discharge between individuals hospitalized before and after the start of the COVID-19 pandemic.

	Admission					Discharge				
Characteristic	Pre-COVID, N = 80 ^1^	Post-COVID, N = 79 ^1^	*p*-Value ^2^	*q*-Value ^3^	ES (95% CI) ^4^	Pre-COVID, N = 80 ^1^	Post-COVID, N = 79 ^1^	*p*-Value ^5^	*q*-Value ^3^	ES (95% CI) ^4^
Age, years	24.88 (8.48)	24.73 (7.80)	>0.9	>0.9	−0.02 (−0.34, 0.30)					
Sex: female	78 (98%)	78 (99%)	>0.9	>0.9						
Diagnosis			0.4	>0.9						
AN-BP	14 (18%)	21 (27%)								
AN-R	47 (59%)	45 (57%)								
ARFID	3 (3.8%)	2 (2.5%)								
BN	6 (7.5%)	7 (8.9%)								
OSFED	10 (12%)	4 (5.1%)								
Duration of Illness, years	6.43 (6.92)	7.30 (7.62)	0.5	>0.9	0.12 (−0.21, 0.44)					
Length of stay, days	34.55 (19.54)	29.08 (18.35)	0.088	0.7	−0.29 (−0.62, 0.05)					
Admission from emergency room	13 (16%)	22 (28%)	0.078	0.7						
BMI (underweight individuals)	14.39 (2.21)	14.30 (1.93)	0.8	>0.9	−0.04 (−0.39, 0.30)	15.08 (2.13)	14.75 (1.82)	0.4	>0.9	−0.17 (−0.53, 0.20)
Caloric intake (underweight), kcal	769.64 (499.88)	786.32 (371.96)	0.8	>0.9	0.04 (−0.33, 0.41)	1457.46 (378.48)	1376.51 (247.12)	0.2	>0.9	−0.25 (−0.62, 0.12)
Physical hyperactivity	24 (30%)	41 (52%)	0.005	**0.050**		8 (10%)	17 (22%)	0.046	0.4	
Binge-purging symptoms	18 (22%)	25 (32%)	0.2	>0.9		2 (2.5%)	2 (2.5%)	>0.9	>0.9	
BSPS fear	9.68 (5.05)	9.61 (6.74)	>0.9	>0.9	−0.01 (−0.34, 0.32)	7.79 (5.01)	8.18 (6.21)	0.7	>0.9	0.07 (−0.27, 0.41)
BSPS avoidance	8.95 (5.23)	9.03 (6.54)	>0.9	>0.9	0.01 (−0.31, 0.34)	7.23 (4.83)	7.85 (6.22)	0.5	>0.9	0.11 (−0.23, 0.46)
BSPS physiologic	3.73 (2.70)	5.53 (5.58)	0.016	0.073	0.41 (0.08, 0.74)	3.06 (2.91)	4.21 (3.62)	0.048	0.4	0.35 (0.01, 0.70)
BSPS total score	22.41 (12.04)	23.49 (16.50)	0.7	>0.9	0.07 (−0.25, 0.40)	17.99 (11.25)	20.21 (14.62)	0.3	>0.9	0.17 (−0.17, 0.52)
EDE-Q restraint	3.14 (2.07)	4.01 (1.86)	0.007	**0.046**	0.44 (0.12, 0.76)	1.75 (1.67)	2.65 (1.57)	0.008	**0.015**	0.56 (0.14, 1.0)
Unknown						29	35			
EDE-Q eating concern	3.01 (1.58)	3.64 (1.35)	0.009	0.054	0.42 (0.10, 0.74)	2.05 (1.39)	2.73 (1.17)	0.011	**0.015**	0.52 (0.11, 0.93)
Unknown						29	35			
EDE-Q shape concern	3.98 (1.58)	4.98 (1.37)	<0.001	**<0.001**	0.67 (0.35, 1.0)	3.01 (1.93)	4.60 (1.40)	<0.001	**<0.001**	0.92 (0.49, 1.3)
Unknown						29	35			
EDE-Q weight concern	3.50 (1.69)	4.49 (1.61)	<0.001	**0.003**	0.60 (0.27, 0.92)	2.46 (1.71)	3.94 (1.58)	<0.001	**<0.001**	0.89 (0.46, 1.3)
Unknown						29	35			
EDE-Q global score	3.41 (1.61)	4.28 (1.38)	<0.001	**0.004**	0.58 (0.26, 0.90)	2.32 (1.52)	3.48 (1.26)	<0.001	**<0.001**	0.82 (0.40, 1.2)
Unknown						29	35			
STAI state anxiety	54.32 (13.05)	60.34 (11.73)	0.004	**0.035**	0.48 (0.15, 0.82)					
STAI trait anxiety	56.90 (15.92)	64.12 (10.01)	0.001	**0.013**	0.54 (0.21, 0.87)					
BDI total score	15.97 (8.73)	19.64 (7.53)	0.015	0.073	0.44 (0.08, 0.81)					

^1^ Mean (SD); n (%); ^2^ Welch Two Sample *t*-test; Fisher’s exact test; Pearson’s Chi-squared test; ^3^ Holm correction for multiple testing. Bold values denote statistical significance at *q* ≤ 0.05; ^4^ Cohen’s d (95% CI). Effects can be interpreted as small for d = 0.2, medium for d = 0.5, and large for d = 0.8; ^5^ Welch Two Sample *t*-test; Pearson’s Chi-squared test; Fisher’s exact test. Abbreviations: AN-R = Anorexia Nervosa—Restricting Type; AN-BP = Anorexia Nervosa Binge-Purging Type; ARFID = Avoidant/Restrictive Food Intake Disorder; BN = Bulimia Nervosa; OSFED = Other Specified Feeding and Eating Disorder; BMI = Body Mass Index; BSPS = Brief Social Phobia Scale; EDE-Q = Eating Disorder Examination Questionnaire; STAI = State-Trait Anxiety Inventory; BDI = Beck Depression Inventory; SD = Standard Deviation; Ci = Confidence Interval; ES = Effect Size.

**Table 2 jcm-12-00573-t002:** Comparison of body specific psychopathological measures between individuals hospitalized before and after the start of the COVID-19 pandemic.

Characteristic	Pre-COVID, N = 53 ^1^	Post-COVID, N = 46 ^1^	*p*-Value ^2^	*q*-Value ^3^	ES (95% CI) ^4^
BCQ specific body parts	20.52 (10.67)	28.89 (10.83)	<0.001	**0.002**	0.77 (0.36, 1.2)
BCQ idiosyncratic checking	9.15 (5.63)	14.83 (6.50)	<0.001	**<0.001**	0.93 (0.51, 1.3)
BCQ total score	56.00 (25.62)	75.76 (25.53)	<0.001	**0.002**	0.77 (0.35, 1.2)
BIAQ clothing	19.13 (9.20)	23.65 (9.43)	0.018	**0.041**	0.48 (0.08, 0.88)
BIAQ social activities	7.58 (5.96)	10.83 (5.43)	0.006	**0.028**	0.56 (0.16, 1.0)
BIAQ eating-related control behavior	5.98 (4.85)	8.33 (4.43)	0.014	**0.041**	0.50 (0.10, 0.90)
BIAQ grooming/weighing	8.72 (3.66)	8.76 (3.40)	>0.9	>0.9	0.01 (−0.38, 0.41)
BIAQ total score	41.42 (18.04)	51.57 (17.47)	0.006	**0.028**	0.57 (0.16, 1.0)
BSQ	114.04 (47.11)	149.35 (40.08)	<0.001	**0.001**	0.80 (0.38, 1.2)

^1^ Mean (SD); ^2^ Welch Two Sample *t*-test; ^3^ Holm correction for multiple testing. Bold values denote statistical significance at *q* ≤ 0.05; ^4^ Cohen’s d (95% CI). Effects can be interpreted as small for d = 0.2, medium for d = 0.5, and large for d = 0.8. Abbreviations: BCQ = Body Checking Questionnaire; BIAQ = Body Image Avoidance Questionnaire; BSQ = Body Shape Questionnaire; SD = Standard Deviation; Ci = Confidence Interval; ES = Effect Size.

**Table 3 jcm-12-00573-t003:** Results from the logistic regression assessing inclusion in pre- and post-pandemic group.

Characteristic	OR ^1^	95% CI ^1^	*p*-Value
Physical hyperactivity	1.95	0.92, 4.19	0.085
EDE-Q restraint	0.76	0.56, 1.03	0.084
EDE-Q shape concern	1.86	1.22, 2.94	0.006
STAI state anxiety	1.00	0.96, 1.04	>0.9
STAI trait anxiety	1.03	0.99, 1.07	0.2

^1^ OR = Odds Ratio, CI = Confidence Interval. Abbreviations: EDE-Q = Eating Disorder Examination Questionnaire; STAI = State-Trait Anxiety Inventory; OR = Odds Ratio; Ci = Confidence Interval.

## Data Availability

Data are not publicly available.

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
