# Peer review of "Increased General, Eating, and Body-Related Psychopathology in Inpatients in a Specialized Eating Disorders Unit after the Beginning of the COVID-19 Pandemic: A Retrospective Comparison with the Pre-Pandemic Period"

_jcm, 2023, doi:10.3390/jcm12020573_

Round 1

Reviewer 1 Report

This study has compared patients with Eating disorders that were treated at an inpatient specialized eating disorder unit, before and after the COVID-19 pandemic. 159 individuals with eating disorders fulfilled the inclusion criteria, 80 before the pandemic and 79 for the two years after March 2020. The authors compared several clinical measures for example on eating disorder psychopathology, anxiety, depression, social phobia and physical activity. The authors find that especially more severe psychopathological scores in most of the areas investigated were found, as well as an increase in physical hyperactivity. They especially stress an increase in body related symptoms for example body shape concerns, body checking and body avoidance. The improvement from treatment during hospitalization were the same between the two groups however, differences in eating symptoms were still evident that this charge.

I find that this is a thoroughly designed, well written, properly organized and well thought through clinical study that is of some relevance to both patients and the scientific community. However, I find that there are some flaws that should be considered. These include the following:

-       albeit that the two populations are similar on several clinical parameters, it cannot be excluded that there are differences between the two populations that is the one included before the pandemic add the one included after the pandemic. The optimal way to conduct this type of investigation would be to include the same patients before and after the pandemic. Although this may have been clinically challenging, and we know that eating disorders change overtime, It would provide the optimal comparison. Any differences found in the current study may always be explained by differences in the two populations. The authors should try and identify patients that were included both before and after the pandemic and make a comparison of their eating disorder of psychopathology and hyperactivity and present this that's one of the main analysis of the current study.

-       The authors find that the group with the most severe eating disorder psychopathology, what's the one group affected the most. This may not be surprising at all since there may be natural causes to explain this finding for example that patience in the highest need of treatment had to wait to be admitted to inpatient treatment due to the pandemic. The authors thereby need to find a way to mitigate for this since this is currently lacking and it's a major flaw.

-       There is a difference in the number of AN-BP patients included pre- and post the pandemic. The authors should weigh in this in the main analyses and also compare AN patients only, also subgroup analyses of balanced groups, since it is AN that is the majority group for inpatient treatment.

Author Response

## reviewer 1

This study has compared patients with Eating disorders that were treated at an inpatient specialized eating disorder unit, before and after the COVID-19 pandemic. 159 individuals with eating disorders fulfilled the inclusion criteria, 80 before the pandemic and 79 for the two years after March 2020. The authors compared several clinical measures for example on eating disorder psychopathology, anxiety, depression, social phobia and physical activity. The authors find that especially more severe psychopathological scores in most of the areas investigated were found, as well as an increase in physical hyperactivity. They especially stress an increase in body related symptoms for example body shape concerns, body checking and body avoidance. The improvement from treatment during hospitalization were the same between the two groups however, differences in eating symptoms were still evident that this charge.

I find that this is a thoroughly designed, well written, properly organized and well thought through clinical study that is of some relevance to both patients and the scientific community. However, I find that there are some flaws that should be considered. These include the following:

thank you for the positive comments

-       albeit that the two populations are similar on several clinical parameters, it cannot be excluded that there are differences between the two populations that is the one included before the pandemic add the one included after the pandemic. The optimal way to conduct this type of investigation would be to include the same patients before and after the pandemic. Although this may have been clinically challenging, and we know that eating disorders change overtime, It would provide the optimal comparison. Any differences found in the current study may always be explained by differences in the two populations. The authors should try and identify patients that were included both before and after the pandemic and make a comparison of their eating disorder of psychopathology and hyperactivity and present this that's one of the main analysis of the current study.

It is a very interesting research idea. However, the number of cases in the databases used for this study is too small to conduct the analyses as suggested. In fact, as we now state in the paper, 12 individuals in the post-COVID group were also present in the pre-COVID group. To account for this, we re-run the analyses on the whole sample after excluding these individuals from the post-COVID group and found analogous results. The table of the comparison can now be found in supplementary materials (supplementary table 1), as now indicated in the paper. We recognize that the differences we found could be in part explained by factors not considered in the study, and we have added this issues to the limitation section:

"The retrospective design warrants caution in interpreting the findings. Notably, we could not isolate the impact of the COVID-19 pandemic from other factors that might have contributed to the differences found between the two separate samples..."

-       The authors find that the group with the most severe eating disorder psychopathology, what's the one group affected the most. This may not be surprising at all since there may be natural causes to explain this finding for example that patience in the highest need of treatment had to wait to be admitted to inpatient treatment due to the pandemic. The authors thereby need to find a way to mitigate for this since this is currently lacking and it's a major flaw.

Following this and the suggestion of reviewer 2 we have modified the discussion of the results regarding the most severe inpatients.

It is true that in order to limit the contagion many patients were only followed at distance (e.g., via phone calls), however, our outpatient service continued to provide in-person appointments when the clinical visit was not  postponeable. We have added these sentences to the limitation section: "...On this line, we cannot infer data on ED incidence, nor on an increased rate of ED individuals in need of hospitalizations in our region. Even if our outpatient service continued to function during the pandemic, the reduction of in-person visits could have contributed to a less prompt response to clinical worsening and therefore to more severe presentations".

-       There is a difference in the number of AN-BP patients included pre- and post the pandemic. The authors should weigh in this in the main analyses and also compare AN  patients only, also subgroup analyses of balanced groups, since it is AN that is the majority group for inpatient treatment.

To account for the issue outlined in this comment we performed supplementary analyses as now explained in the paper: "The group of individuals hospitalized after March 2020 showed an increase in percentage of individuals diagnosed with AN-BP (table 1), even though there were no significant differences in the comparison of all the diagnoses. To account for potential effects of subgroup imbalances, we conducted factorial ANOVAs on the population diagnosed with AN, using presence of hyperactivity, EDE-Q and STAI subscales as dependent variables, and the period of hospitalization (before and after COVID-19), the diagnostic sub-type (AN-R an AN-BP), and the interaction between the two as fixed factors (supplementary table 2). For all variables considered there was an effect of the period of hospitalization, being the post-COVID period associated with significantly higher scores. For EDE-Q restraint, eating concern, weight concern and global score there was also an effect of diagnostic subtype, being AN-BP associated with higher scores. For no variables there was a significant effect of the interaction between period of hospitalization and diagnostic subtype."

Reviewer 2 Report

Thank you for the opportunity to review this important manuscript, which aims to characterize and assess, via retrospective analysis, differences in clinical presentation, psychopathology, and treatment outcomes of adult inpatients with eating disorders in the two years before vs. after the start of the COVID-19 pandemic. This topic is important and the paper is generally well-written. However, as I outline below, there are a number of places where the authors could provide additional detail or clarification that would further strengthen the paper.

Introduction

Overall the introduction does a nice job of introducing the study. However, adding some details about studies mentioned in the literature review would help the reader further contextualize the present study: 

1) Lines 39-41: Can you clarify that the Castellini study was a prospective longitudinal study that assessed patients before and during the pandemic? Reading this sentence, I was not sure if this was a cross-sectional study and whether it included pre-pandemic assessment. I think it would be helpful to clarify.

2) Lines 65-72: Can you clarify that rates of hospital admission for EDs were generally stable prior to the pandemic, so that it is clear that this is a pandemic-related increase in admissions (not a general trend that was already occurring prior to the pandemic).

3) Lines 75-78: Can you please also state the Schreyer study's findings regarding anxiety and discharge outcomes? I think it is relevant to note since your study also aim to assess these factors.

Aims:

1) To clarify - is it correct that Aims A, B, and secondary Aim A have already been examined in prior studies, or is there something different/new about these aims? It is fine if they are essentially replication aims, but it would be helpful to clarify somewhere so the reader knows what is new versus attempting to replicate/build on prior data. And, is it also correct that Aim C and Secondary Aim B are novel? Again, would be helpful to make it super clear to readers in the introduction what is brand new, and why it is also helpful to examine questions that have already been examined (could just be because the data are limited). 

Method

1) Why were adults >65 years excluded?

2) Were participants with psychotic, bipolar, or active substance use disorders excluded from this study, or were they excluded from treatment at the center and treated elsewhere due to comorbidities? Just trying to understand if data on these participants were excluded or if participants with these characteristics were simply not treated at the Eating Disorder Center.

3) Measures: It is currently unclear where each of the measures comes from. For example, I can infer that the diagnosis comes from the SCID (which you mention in the previous section) and that presence of binge-purging symptoms either comes from the SCID or the EDE-Q, but it would be very helpful to specify/provide more detail about where each of the measures comes from/how it was assessed. For example, how was "presence of excessive physical activity" operationalized and assessed? What about caloric intake at admission and discharge? Was duration of illness self-reported? 

4) Line 147: Can you specify how many of the n=99 were from the pre- versus post-pandemic groups? This is included in a later table, but would be helpful to specify here. Why was this measure missing for so many more people, compared to the other measures? 

5) Lines 161-169: Given that lockdown/reopening procedures differed country to country, it would be helpful to briefly define what "lockdown" and "phase of movement restrictions" meant in Italy. 

6) Did you assess any demographic characteristics besides age and sex? Gender? Sexual orientation? Race? Ethnicity? Income or SES? Etc. These variables can help contextualize findings and speak to generalizability even if they are not central to your research question.

Results

1) You later state in the Discussion that the "two groups [showed] a comparable reduction in eating symptomatology during the inpatient stay" (lines 267-268). Did you statistically examine if this was the case? That is, did you look at whether change scores from admission to discharge statistically differed by group? 

2) Can you state if any of the lockdown/reopening comparisons for BMI and EDE-Q are significant?

3) Section 3.4- Extreme-AN: Can you clearly state in this section the n's for extreme AN?

Discussion

1) Lines 322-323: You state that results suggest the pandemic ESPECIALLY affected those with extreme-AN. Can we state this for sure based on findings, or is it possible that the smaller sample size of those with extreme-AN could be influencing effect sizes? 

Author Response

## reviewer 2

Thank you for the opportunity to review this important manuscript, which aims to characterize and assess, via retrospective analysis, differences in clinical presentation, psychopathology, and treatment outcomes of adult inpatients with eating disorders in the two years before vs. after the start of the COVID-19 pandemic. This topic is important and the paper is generally well-written. However, as I outline below, there are a number of places where the authors could provide additional detail or clarification that would further strengthen the paper.

Thank you for the positive comments and the careful reading.

**Introduction**

Overall the introduction does a nice job of introducing the study. However, adding some details about studies mentioned in the literature review would help the reader further contextualize the present study: 

1) Lines 39-41: Can you clarify that the Castellini study was a prospective longitudinal study that assessed patients before and during the pandemic? Reading this sentence, I was not sure if this was a cross-sectional study and whether it included pre-pandemic assessment. I think it would be helpful to clarify.

We have modified the description of the study in this way: "Castellini and colleagues [12] examined outpatients with an ED both before the pandemic and in the first months after its inception and compared them to healthy controls, finding..."

2) Lines 65-72: Can you clarify that rates of hospital admission for EDs were generally stable prior to the pandemic, so that it is clear that this is a pandemic-related increase in admissions (not a general trend that was already occurring prior to the pandemic).

The modified sentence reads as follows: "Regarding inpatient ED treatment, an increase in the percentage of hospitalizations has been reported in children [17] and adolescents [18], whereas hospitalization rate for this age group was generally stable in the 5 years before the pandemic [19]."

3) Lines 75-78: Can you please also state the Schreyer study's findings regarding anxiety and discharge outcomes? I think it is relevant to note since your study also aim to assess these factors.

Thank you for pointing this out, we have addedd the following sentence in the description of Schreyer et al work: "For both age groups, no significant differences were found regarding anxiety levels at admission and weight outcome at discharge".

**Aims:**

1) To clarify - is it correct that Aims A, B, and secondary Aim A have already been examined in prior studies, or is there something different/new about these aims? It is fine if they are essentially replication aims, but it would be helpful to clarify somewhere so the reader knows what is new versus attempting to replicate/build on prior data. And, is it also correct that Aim C and Secondary Aim B are novel? Again, would be helpful to make it super clear to readers in the introduction what is brand new, and why it is also helpful to examine questions that have already been examined (could just be because the data are limited). 

That is correct. At the time we started the analyses most of the aims were novel, since there was a lack of data on adult inpatients. However, especially after the paper by Schreyer et al. some of them have been explored, albeit with a different focus (i.e., comparison between adult and adolescents).

We now better clarify in the Aims sections: "Aim a) and b) have been explored in the adult ED inpatient population by Schreyer et al. (2022), albeit with a different focus (i.e., comparison between adolescent and adult populations), whereas aim c) is novel. Similarly, secondary aim a) has been explored in the ED population attending specialized ED centers [13], whereas we are not aware of studies assessing secondary aim b)."

**Method**

1) Why were adults >65 years excluded?

Elderly individuals are usually excluded in order to provide a sample that is representative of the typical demographics of individuals with eating disorders.

2) Were participants with psychotic, bipolar, or active substance use disorders excluded from this study, or were they excluded from treatment at the center and treated elsewhere due to comorbidities? Just trying to understand if data on these participants were excluded or if participants with these characteristics were simply not treated at the Eating Disorder Center.

Individuals who met the above-mentioned exclusion criteria were treated at the center but were not included in the study, due to alterations in cognition associated with either acute or chronic psychosis and/or substance effects.

3) Measures: It is currently unclear where each of the measures comes from. For example, I can infer that the diagnosis comes from the SCID (which you mention in the previous section) and that presence of binge-purging symptoms either comes from the SCID or the EDE-Q, but it would be very helpful to specify/provide more detail about where each of the measures comes from/how it was assessed. For example, how was "presence of excessive physical activity" operationalized and assessed? What about caloric intake at admission and discharge? Was duration of illness self-reported? 

We expanded the Measures section with this paragraph:  " Clinical measures were collected via a form filled out by the clinician after the admission visit and at discharge. Duration of illness was either known for individuals who attended the Center in other settings or reconstructed via patient history collection. Similarly, presence of binge-purging symptoms and excessive physical hyperactivity were both operationalized as dichotomous variables indicating whether these symptoms were present at a clinical level in the month prior the assessment at admission; their presence during the inpatient stay was assessed via daily clinical observation and visits. Caloric intake at admission was estimated through self-report with the help of the nutritionist that then followed the patient during the hospital stay. "

4) Line 147: Can you specify how many of the n=99 were from the pre- versus post-pandemic groups? This is included in a later table, but would be helpful to specify here. Why was this measure missing for so many more people, compared to the other measures? 

Those questionnaires are part of an expanded version of the baseline assessment, which is usually reserved for inpatients that have not completed them elsewhere at the ED-center. These measures could be missing from the database due to one of the following reasons: 1) the individual did not receive them, e.g. because she/he had already recently completed them in the outpatient setting, 2) the individual received the complete assessment but completed only the questionnaires reported in table 1, 3) data could not be retrieved due to errors in calculation or reporting in the database. We are currently unable to provide percentages for these points, however, we have no reason to suspect that the subpopulation for which these measures were available constituted a clinically different population from the whole sample.

5) Lines 161-169: Given that lockdown/reopening procedures differed country to country, it would be helpful to briefly define what "lockdown" and "phase of movement restrictions" meant in Italy. 

We have expanded the sentence as advised: "Lockdown in Italy entailed nationwide quarantine and the closure of all non-essential business and were put in place in the following periods: first lockdown between March 2020 and June 2020; second lockdown between October 2020 and the end of April 2021; a phase of movement restrictions (i.e., the enduring of the state of emergency with limitations in the access to non-essential business) between December 2021 and the end of March 2022."

6) Did you assess any demographic characteristics besides age and sex? Gender? Sexual orientation? Race? Ethnicity? Income or SES? Etc. These variables can help contextualize findings and speak to generalizability even if they are not central to your research question.

Most of the above-mentioned measures were collected, however, were not included in the present analysis to limit the number of variables compared. As we now state in the paper, "The ED Center is a secondary treatment center fully founded by the public Italian healthcare system, and access to care is free (i.e., not based on insurances). All individuals treated in the period considered were non-Hispanic White." Information on sexual orientation and gender identity is not routinely reported for clinical studies since it is regarded as highly sensitive by our Ethical Committee.

We have added these issues to the limitation section: "The findings of the present study were obtained from White individuals therefore lack of generalizability to individuals of more diverse racial an ethnic background. Furthermore, we did not report information on sexual orientation and gender identity since it was not consistently collected."

**Results**

1) You later state in the Discussion that the "two groups [showed] a comparable reduction in eating symptomatology during the inpatient stay" (lines 267-268). Did you statistically examine if this was the case? That is, did you look at whether change scores from admission to discharge statistically differed by group? 

We now report in the Results section that the "delta in EDE-Q global scores did not differ in the two samples (t(89.59) = -0.54, p = 0.592)".

2) Can you state if any of the lockdown/reopening comparisons for BMI and EDE-Q are significant?

We have modified the lockdown paragraph in this way: "The comparison between inpatients hospitalized during lockdowns and during the re-opening phases did not show differences in BMI nor EDE-Q subscales. Initially, significantly higher scores in physiological symptoms of social anxiety reported in the BSPS interview emerged at discharge for the re-opening group (supplementary table 1), however, after controlling for multiple comparisons no significant differences survived".

3) Section 3.4- Extreme-AN: Can you clearly state in this section the n's for extreme AN?

"Regarding general clinical measures and questionnaires on the 74 inpatients with extreme-AN (34 pre-COVID, 39 post-COVID),..."

**Discussion**

1) Lines 322-323: You state that results suggest the pandemic ESPECIALLY affected those with extreme-AN. Can we state this for sure based on findings, or is it possible that the smaller sample size of those with extreme-AN could be influencing effect sizes?

We have rephrased the sentence at the end of the first paragraph of the discussion: "Differences in specific eating and body-related psychopathology were also evident for the subgroup of inpatients with extreme-AN." Similarly, another passage now says: "The separate analysis of individuals with extreme-AN suggested that also this population was affected by the pandemic." And in limitations: "the number of individuals in the subgroup analyses was limited, especially for individuals with extreme-AN."

Round 2

Reviewer 1 Report

I find that the authors have responded to my comments. An important addition to be made is to include the word "retrospective" in the title and in the abstract.

1) The title should thereby read:

"Retrospective analysis of clinical aspects of Inpatients in a specialized eating disorders unit before and after the beginning of the COVID-19 pandemic: Increase in general, eating, and body-related psychopathology"

Should the title be too long, a rewording including the word "retrospective" is mandatory.

2) This should also be visibly included in the abstract, bot the design and the interpretation. This sentence, as proposed by the authors "Notably, we could not isolate the impact of the COVID-19 pandemic from other factors that might have contributed to the differences found between the two separate samples..." should also be invcluded in the abstract, fitted to the number of words required.

Author Response

I find that the authors have responded to my comments. An important addition to be made is to include the word "retrospective" in the title and in the abstract.

Thank you, we have made changes to the abstract and the title as suggested.

1) The title should thereby read:

"Retrospective analysis of clinical aspects of Inpatients in a specialized eating disorders unit before and after the beginning of the COVID-19 pandemic: Increase in general, eating, and body-related psychopathology"

Should the title be too long, a rewording including the word "retrospective" is mandatory.

We have modified the title in this way: "Increased general, eating, and body-related psychopathology in inpatients in a specialized eating disorders unit after the beginning of the COVID-19 pandemic: A retrospective comparison with the pre-pandemic period"

2) This should also be visibly included in the abstract, bot the design and the interpretation. This sentence, as proposed by the authors "Notably, we could not isolate the impact of the COVID-19 pandemic from other factors that might have contributed to the differences found between the two separate samples..." should also be invcluded in the abstract, fitted to the number of words required.

We have incorporated the sentence as suggested: "This retrospective analysis does not allow us to separate the impact of COVID-19 from other potentially relevant co-occurring factors, however, these findings help in understanding how the pandemic could have affected individuals that needed specialized intensive treatment."